# Locally Isolated *Trichoderma harzianum* Species Have Broad Spectrum Biocontrol Activities against the Wood Rot Fungal Species through Both Volatile Inhibition and Mycoparasitism

**DOI:** 10.3390/jof9060675

**Published:** 2023-06-15

**Authors:** Mu En Chan, Jhing Yein Tan, Yan Yi Lee, Daryl Lee, Yok King Fong, Marek Mutwil, Jia Yih Wong, Yan Hong

**Affiliations:** 1School of Biological Sciences, Nanyang Technological University, 60 Nanyang Drive, Singapore 637551, Singapore; chan0938@e.ntu.edu.sg (M.E.C.); jhingyein.tan@ntu.edu.sg (J.Y.T.); ylee083@e.ntu.edu.sg (Y.Y.L.); mutwil@ntu.edu.sg (M.M.); 2National Parks Board, 1 Cluny Road, Singapore Botanic Gardens, Singapore 259569, Singapore; daryl_lee@nparks.gov.sg (D.L.); fongyokking168168@yahoo.com (Y.K.F.); wong_jia_yih@nparks.gov.sg (J.Y.W.)

**Keywords:** biocontrol, *Trichoderma*, volatile inhibition, mycoparasitism, wood rot fungi, *Phellinus noxius*, *Rigidoporus microporus*, *Fulvifomes siamensis*

## Abstract

Pathogenic root/wood rot fungal species infect multiple urban tree species in Singapore. There is a need for sustainable and environmentally friendly mitigation. We report the local *Trichoderma* strains as potential biocontrol agents (BCAs) for pathogenic wood rot fungal species such as *Phellinus noxius*, *Rigidoporus microporus*, and *Fulvifomes siamensis*. Isolated *Trichoderma* strains were DNA-barcoded for their molecular identities and assessed for their potential as a BCA by their rate of growth in culture and effectiveness in inhibiting the pathogenic fungi in in vitro dual culture assays. *Trichoderma harzianum* strain CE92 was the most effective in inhibiting the growth of the pathogenic fungi tested. Preliminary results suggested both volatile organic compound (VOC) production and direct hyphal contact contributed to inhibition. SPME GC-MS identified known fungal inhibitory volatiles. *Trichoderma harzianum* strain CE92 hyphae were found to coil around *Phellinus noxius* and *Lasiodiplodia theobromae* upon contact in vitro and were possibly a part of the mycoparasitism. In summary, the work provides insight into *Trichoderma* inhibition of pathogenic fungi and identifies local strains with good potential for broad-spectrum BCAs against root/wood rot fungi in Singapore.

## 1. Introduction

Soil-borne plant pathogenic fungi are environmentally ubiquitous and represent a significant causative agent of root rot in plants [1]. These fungi are able to break down the key wood structural components through enzymatic degradation, and commonly infect plants through their roots but may also attack the stem bases and branches. Being soil-borne, infection by pathogenic fungi tends to start from root systems that are underground and hidden from sight, causing difficulty in early detection [2]. Such fungal infections may become difficult to treat by the time the infection is discovered through notable late-developing symptoms, such as crown dieback and fruiting body formation [3]. These pathogenic fungi have also been found to persist within the soil for extended periods of time even after the death and removal of the formerly infected tree and thus can infect other trees in the vicinity [4]. Infected roots hamper the ability of the trees to absorb water and nutrients from their surroundings. Common symptoms may range from the softening of root tips and root lesions to crown wilting, leading to compromised growth and, eventually, death [5].

As part of the City in Nature vision for Singapore, the maintenance of local urban forestry, including common species such as *Samanea saman* (Rain tree), *Peltophorum pterocarpum* (Yellow Flame), *Pterocarpus indicus* (Angsana), and *Khaya senegalensis* (Khaya Tree) is critical. Root rot is caused by a pathogenic fungal infection of urban trees, leading to compromised structural integrity, and can be of key concern. Based on a metagenomic survey of pathogenic wood rot fungi in Singapore [6], several species of root rot fungi, including *Fulvifomes siamensis, Phellinus noxius*, and *Rigidoporus microporus*, have been observed to infect multiple species of local urban trees, including Rain Tree, Angsana (*Pterocarpus indicus*), *Casuarina equisetifolia* (Casuarina), Yellow Flame, and Khaya Tree. The infection of urban trees can pose safety risks to life and property should the continuous infections cause tree failures. As such, developing methods for early detection of diseased trees and inhibition of pathogenic fungi growth in local tree species remain relevant.

While there are agrochemicals available to address pathogenic fungal growth, there is interest in developing more environmentally friendly alternatives, such as biocontrol agents (BCAs), which involve using natural or modified fungal species to target pathogenic species [7]. Benefits of using BCAs include higher target specificity and adaptability and lower maintenance requirements due to the self-sustaining nature of BCAs. Currently, the *Trichoderma* species appear to be promising as a BCA targeting and inhibiting a wide range of fungal phytopathogens [8].

*Trichoderma* species are ubiquitous in most soil environments and have been found to antagonise other fungal species through mechanisms such as mycoparasitism, antibiosis, and resource competition [9]. *Trichoderma* species are also opportunistic, avirulent plant symbionts that readily colonise plant roots and enhance the growth of their host plants [10]. These traits have made *Trichoderma* species a key fungus in antifungal BCA-related research. Much research has been conducted to analyse their ability to mitigate damage caused by pathogenic fungal species of crop plants, such as soybean [11] and maize [12]. There are also reports of *Trichoderma* species used to control various wood rot fungal species, such as *Ganoderma* sp. [13,14,15,16], *Rigidoporus microporus* [17], *Phellinus noxius* [18], and *Lasiodiplodia theobromae* [19,20]. Two *Trichoderma* species were tested in preventing wood decay after inoculation into pruning wounds of Benin mahogany and Rain Tree [21] in Singapore. Most of these reports explored local *Trichoderma* isolates against a single pathogenic fungal species, and there was no report on the biocontrol of *F. siamensis*, the white wood rot that infected multiple urban trees in Singapore.

Several *Trichoderma* species, including multiple strains of *Trichoderma harzianum*, have been isolated in this study and prior studies. One objective of this study is to evaluate the potential of local *Trichoderma harzianum* strains to combat the major root rot fungi in Singapore and to identify a strain with the most potential for field application as a BCA. As a result, a local *Trichoderma harzianum* strain was found to have strong and broad-spectrum in vitro inhibition activities on the major root/wood fungi in Singapore.

The second objective of the study is to understand the mechanisms underlying the inhibitory effects that *Trichoderma* species employ against pathogenic fungal species through three tests: (1) inhibition by volatiles followed by their chemical structure characterization by SPME GC-MS, (2) inhibition by soluble secretions, and (3) microscopic observation of the interface between *Trichoderma* sp. and pathogenic fungi. It was found that the *Trichoderma* strain could inhibit the wood rot fungi through long-range, volatile compounds, as well as through direct hyphal contact and interactions. The hyphae of the *Trichoderma harzianum* strain CE92 were found to coil around the hyphae of *Phellinus noxius* and *Lasidioplodia theobromae* isolates, thus suggesting mycoparasitism.

## 2. Materials and Methods

### 2.1. Culture Media Preparation

The medium used in this study was potato dextrose agar (PDA). PDA was prepared using PDA powder (Sigma Aldrich, St. Louis, MO, USA), autoclaved at 121 °C for 15 min, and cooled to approximately 40 °C before the addition of streptomycin and ampicillin to inhibit bacterial growth. Quantities of components used in medium preparation are detailed in Table 1. A 20 mL media solution was poured into each 90 × 15 mm polystyrene petri dish (Biomedia, Singapore) and stored at 4 °C.

### 2.2. Culturing and Isolation of Fungi

Diseased tissue and fruiting body samples were cut into small fragments by a sterile hand saw that was disinfected with 70% ethanol. The fragments were placed equally spaced apart onto PDA with antibiotics (streptomycin 30 mg/L + ampicillin 100 mg/L). Soil samples were suspended with autoclaved water, filtered, and diluted 100× before inoculation. Fungal colonies growing from the fragments or soil samples were considered as the 1st generation (G1) isolates. Colonies were sub-cultured onto PDA plates with antibiotics to obtain pure cultures. Sub-culturing for pure isolates was performed by transferring hyphae from the edge of the fungal colony to a new culture plate with sterile scalpel blades. All plates were incubated at 30 °C in the dark to promote fungal growth.

Culturing for experiments involved cutting mycelial plugs from pure culture plates using a scalpel sterilised in 100% ethanol and flamed over a spirit lamp before the transfer of plugs to a PDA plate for incubation at room temperature. All culturing (date, medium used, and generation number) was recorded clearly on each plate. DNA extraction and sequencing were conducted to confirm the purity of the culture and fungal species prior to the start of any experiments.

### 2.3. DNA Extraction from Fungal Cultures

DNA extraction from fungal mycelia was conducted following a Lyticase-Chelex 100 DNA isolation protocol adapted from Pryce et al. [22] with some modifications. Approximately 2–4 mm^2^ mycelia mass was scraped into a sterile 1.5 mL microcentrifuge tube. An extraction buffer was prepared by mixing 180 µL 10 mM Tris-HCl pH 8.0 with 20 µL 1.0 unit/mL Lyticase stock solution (Sigma-Aldrich, Singapore). The extraction buffer was added to mycelia and vortexed for 30 s before incubation at 37 °C for 45 min followed by vortexing for 30 s. A total of 20mg of Chelex^®^ 100 (Bio-Rad Laboratories, Hercules, CA, USA) was added per tube before being vortexed for 30 s and incubated at 95 °C for 10 min. Samples were cooled to room temperature before centrifugation at 13,000× *g* for 2 min. The supernatant was transferred to sterile 1.5 mL microcentrifuge tubes for DNA purification using an EZ-10 Spin Column PCR Products Purification Kit (BioBasic Asia Pacific, Singapore) following the manufacturer’s protocol. A total of 40 µL of sterile water was used to elute the DNA. The concentration and purity of eluted DNA were determined using Nanodrop 1000 (Thermo Scientific, Waltham, MA, USA). Eluted DNA was stored at −20 °C for further use. For fungal species with tougher mycelia, including *Fulvifomes siamensis*, DNA extraction was conducted using the DNeasy^®^ PowerLyzer^®^ PowerSoil^®^ Kit (Qiagen, Hilden, Germany), according to the manufacturer’s protocols, before purification using an EZ-10 Spin Column PCR Products Purification Kit.

### 2.4. Barcoding for Fungal Isolates

Barcoding for fungal isolates was conducted by following a standard procedure [6]. Briefly, DNAs from isolates were PCR-amplified for the ITS1-5.8S-ITS2 region with the V9D and LS266 primer pair (5′-TTAAGTCCCTGCCCTTTGTA-3′ and 5′-GCATTCCCAAACAACTCGACTC-3′, respectively) with the following program: initial denaturation at 95 °C for 5 min; followed by 35 cycles of 30 s denaturation at 95 °C, 30 s annealing at 50 °C, and 30 s extension at 72 °C; and a final extension of 72 °C for 5 min. PCR amplicons were purified with the QiaQuick PCR purification kit (Qiagen, Hilden, Germany) prior to submission for sequencing (Bio Basic Asia Pacific Pte Ltd., Singapore) with the primers ITS5 and ITS2 (5′-GGAAGTAAAAGTCGTAACAAGG-3′ and 5′-GCTGCGTTCTTCATCGATGC-3′, respectively). Sequence chromatograms were visualised for verification/manual correction, and consensus sequences were assembled by Geneious Prime (2 February 2022). Geneious Prime was used to conduct a Basic Local Alignment Search Tool (BLAST) search in the “Nucleotide collection (nr/nt) database” for non-redundant GenBank entries to determine the identity of cultured fungal isolate based on the best match (lowest e value that is less than the cut-off value of 1e-100) with full annotation down to the species level. Consensus sequences and phylogenetic trees were also generated to help in the selection of *Trichoderma* strains to be used for experiments.

### 2.5. Growth Rate Comparison among the Trichoderma Strains

Growth rate comparisons were conducted on seven pure *T. harzianum strains* to determine the fastest-growing strains. Five replicates were used for each *Trichoderma* strain. *Trichoderma* plugs approximately 0.9 cm in diameter were cut using the base of a P1000 micropipette tip and inoculated onto the edge of a PDA plate. Radial growth of *Trichoderma* was recorded daily until growth had reached the opposite end of the plate. The growth rate for each sample was determined as the slope for the linear regression between the time (independent variable) and growth (dependent variable) by Microsoft Excel. The correlation coefficient was also calculated to confirm the linear relationship between time and growth.

### 2.6. Dual Culture Assay

A dual culture assay was adapted from a protocol by Singh et al. [23]. Six pathogenic fungi were plated against three selected strains of *T. harzianum*. Fungal plugs were cut from pure culture fungal plates using the base of a P1000 micropipette tip. For fast-growing pathogenic fungal species, *Phellinus noxius*, *Rigidoporus microporus* and *Lasiodiplodia theobromae*, experimental plates were inoculated by one pathogenic fungal plug on one end and one *Trichoderma* plug on the opposite end of the PDA plate at the same time. For the slower-growing fungal species, *Fulvifomes siamensis*, *Ganoderma orbiforme*, and *Ganoderma australe*, inoculating *Trichoderma* plugs opposite the pathogenic fungal plug was conducted after radial growth of pathogenic fungi had reached approximately halfway through the plate, usually one week later. Each assay had triplicates. All dual culture assay plates were incubated at 24 °C, and radial growth of mycelia of both fungi growing on the plate was recorded every 24 h over a period of seven days.

### 2.7. Inhibition Test by Volatile Organic Compound (VOC)

The VOC inhibition test was adapted from two previous reports [24,25]. Experimental plates were prepared by placing a pathogenic fungal plug at the center of a PDA plate and a *T. harzianum* plug at the center of a separate PDA plate. The plate inoculated with the pathogenic fungus was inverted and placed over the *Trichoderma* plate. The rims of both plates were aligned and sealed. Plates were incubated at 24 °C, and the diameter of pathogenic fungal mycelia was recorded every 24 h for seven days. The control plates were similarly prepared with a clean PDA plate replacing the *Trichoderma* plate.

### 2.8. SPME GC-MS (Solid Phase Microextraction Gas Chromatography–Mass Spectrometry)

VOCs generated by pathogenic fungi and *Trichoderma* in the VOC test were investigated using SPME GC-MS analysis according to Hong et al. [6]. Briefly, samples were prepared by placing six cut plugs into 20 mL SPME clear rounded bottom vials crimped using 20 mm Black Viton^®^ Septa Seals (Supelco Inc., Bellefonte, PA, USA). An SPME Fiber Assembly 50/30 µm Divinylbenzene/Carboxen/Polydimethylsiloxane (DVB/CAR/PDMS) (Supelco Inc., Bellefonte, PA, USA) was inserted through the septa to capture headspace volatiles over two days at room temperature. The injection was performed in a splitless mode. The chromatographic analysis for each sample was performed in an Agilent 7890A/5975 GC-MS system using Agilent J&W HP-5ms 30 m × 0.25 mm × 0.25 µm GC column (19091S-433) for 50 min. The initial oven temperature was kept isocratic at 40 °C for 2 min, before being increased by 6 °C/min to 80 °C, followed by an increase of 3.4 °C/min to 170 °C, before being increased by 12 °C/min to a peak temperature of 300 °C, which was maintained for 4 min. The MS detector was working at a mass range of 40–500 amu at positive polarity. An MSD ChemStation (Agilent Technologies, Santa Clara, CA, USA) and its default analytical software was utilised to generate a volatile emission profile, with compound identification in comparison to the MIST Mass Spectral Library.

### 2.9. Microscopy Analysis of Fungi Interface

Based on Cheng et al. [26] who reported an antagonistic assay visualised under a light microscope, 1mL of unsolidified PDA medium was spread thinly over a sterile Citoglas^®^ Super Grade 25 × 75 mm microscope slide. Hyphal tips of a *T. harzianum* strain and a pathogenic fungus were inoculated onto opposite ends of the PDA-coated microscope slide. Microscope slides were sealed in a Petri dish and incubated at room temperature and observed every day until the mycelia of both fungi intersected. Lactophenol cotton blue solution (Sigma-Aldrich, St. Louis, MO, USA) was used to stain fungal mycelia. The slide was viewed under a Leica DM6000B Light Microscope equipped with a 40× objective lens.

### 2.10. Statistical Analysis

All results involving triplicates or five replicates are reported in the form of mean ± standard deviation (SD). Tukey’s honestly significant difference (HSD) test was used to compare the growth rates of the seven *T. harzianum* strains and determine the variation among the different strains of *T. harzianum* in inhibition of the pathogenic fungal strains at a 95% significance level (*p* < 0.05) using the Real Statistics Resource Pack add-in for Microsoft Excel. Compact Letter Display was used to display statistically significant different categories. Letter grades were assigned in alphabetical order, with “a” suggesting the highest inhibitory potential and subsequent letters indicating a statistically significant lower inhibitory potential.

## 3. Results

### 3.1. Identities of Fungal Isolates Were Elucidated by DNA Barcoding

A total of seven *T. harzianum* strains and six pathogenic fungal species were included in this study (Table 2). Their molecular identities were determined by barcoding the ITS1-5.8S-ITS2 region.

*R. microporus*, is a member of the order *Polyporales*, and *F. siamensis* and *P. noxius* are members of the order *Hymenochaetales*. These three pathogenic white rot fungi were identified as possible major threats to urban trees in Singapore [6], and their isolates were included in this study (P175, H-FS and KS71, coded H-Rm, H-Fs and H-Pn respectively). *Ganoderma* sp. was found not only infecting palms but also urban trees, such as Casuarina; hence, one *G. orbiforme* (P167, coded H-Go) and one isolate of *G. australe* (P172, coded H-Ga) were included in this study. *L. theobromae*, a *Botryosphaeriales* fungus identified to cause root rot and collar rot disease reported in crop trees, such as *Jatropha curcas* [27] and Mulberry [28], was isolated from diseased tissues of two urban tree species, Purple Millittia and Khaya, and also included in this study (PM162, coded H-Lt, Table 2).

For the multiple *Trichoderma* isolates from different sources, we used Geneious Prime software to conduct a global alignment with 93% similarity of the sequence data from all seven pure strains of our locally isolated *T. harzianum*, together with various *Trichoderma* sequences obtained from GenBank, with their accession numbers indicated in Figure 1. These *Trichoderma* strains obtained from GenBank are well-characterised and were included in the phylogenetic tree as reference points. The neighbour-joining phylogenetic tree was then constructed using Geneious Prime software with a Tamura-Nei distance model and 100 bootstrap values (Figure 1). The tree was rooted with the outgroup taxa *Trichoderma reesei* (accession number OW985639), *Trichoderma viride* (accession number MF432726), and *Trichoderma hamatum* (KU196764). Based on this phylogenetic analysis, all the strains belong to *Trichoderma harzianum* and are closely related to each other.

### 3.2. T. harzianum Strains CE92, SSK1, and W2-5 Selected for Further Study

Figure 2 summarises the results of the *T. harzianum* growth comparison. The growth of all seven strains followed a near-perfect positive linear relationship with time (with a correlation coefficient >99.0 for all samples). Of the seven isolates of *T. harzianum*, CE92 and W2-5 displayed the fastest radial growth, covering the entirety of the plate by Day 5.

Based on the growth rates of the tested *T. harzianum* strains, CE92 and W2-5 (coded Th-1 and Th-3 hereafter) were chosen for this study for the fastest radial growth among all the *Trichoderma* strains. A third strain SSK1 was chosen (coded Th-2 hereafter) for being fast-growing with dense white mycelia that appeared furry, similar to Th-1. Th-3 was observed to have a major morphology different from Th-1 and Th-2, as it had mycelia that were smooth on the surface and penetrated deeper into the agar surface with a vein-like pattern (Figure 3).

### 3.3. Th-1 and Th-2 Displayed Potent Inhibitory Activity against Pathogenic Fungi in a Dual Culture Assay

A dual culture assay was conducted using the three shortlisted *T. harzianum* strains against six pathogenic fungal species. Percentage inhibition of pathogenic fungi by *Trichoderma* was calculated according to:$$\mathrm{Inhibition}\;\mathrm{percentage}=\frac{R_{C}-R_{E}}{R_{C}}\times 100$$
where R_C_ is the radial growth of pathogenic fungi in the control plate and R_E_ is the radial growth of pathogenic fungi in experiment plates on Day 7.

Pathogenic fungi of interest could be separated into fast growers and slow growers. Fast growers comprised *Phellinus noxius* (H-Pn), *L. theobromae* (H-Lt), and *R. microporus* (H-Rm), while slow growers comprised *G. orbiforme* (H-Go), *G. australe* (H-Ga), and *F. siamensis* (H-Fs). Analysis of the dual culture assay was mainly conducted using data from simultaneous inoculation of pathogenic fungi and *Trichoderma* for the fast growers, while data from experiments where pathogenic fungi were inoculated first was used for the slow growers.

Figure 4 shows the representative plates for the dual culture assays and Figure 5 quantitatively compared the percentage of inhibition efficiency.

*P. noxius* (H-Pn) in the control plates was observed to produce fine white mycelia, with radial growth rate increasing exponentially after Day 3 and reaching a radial growth of 7.8 cm on Day 7. Little or no growth for H-Pn mycelia was observed in dual culture assays with Th-1 and Th-2 but some growth was observed with Th-3. Quantitatively, Th-1 and Th-2 were equally efficient and slightly better than Th-3, but not significantly different based on Tukey’s test.

A dual culture assay where H-Pn was inoculated before the *Trichoderma* strains was also conducted. In all plates, similar inhibition trends were observed compared to simultaneous inoculation.

*R. microporus* (H-Rm) was observed to produce a steady growth of white, heavily branched mycelia, which reached a radial growth of 4.7 cm on Day 7 in the control plate. In the presence of *Trichoderma* strains, stagnation in radial growth was observed for all strains of *Trichoderma* (Figure 4). While the intersection boundary was maintained, H-Rm appeared to adopt a more defensive mycelial structure that resembled a ball with growth only within the fungal colony. Quantitatively, all three *Trichoderma* strains had similar percentage inhibition (Figure 5). A dual culture where H-Rm was inoculated before the *Trichoderma* strains was also conducted. Similar growth stagnation after inoculation of *Trichoderma* strains was observed. The percentage inhibition caused by the three *Trichoderma* strains was not significantly different from each other based on Tukey’s test.

*L. theobromae* (H-Lt) is a non-host-specific opportunistic fungal pathogen belonging to *Botryosphaeriales* and is known to cause black root rot disease where infections start as black lesions in the middle of roots and subsequently expand to form cankers [27,28,29]. Apart from a strain isolated from the Purple Millettia tree (H-Lt), it was also isolated from the decayed wood tissues of the Khaya tree (GenBank No. OQ558856). H-Lt was observed to produce rapid growth of thick white mycelia, which gradually turned black over time, fully covering the control plate in mycelia by Day 5, which corresponded to a radial growth of 8.6 cm. After inoculation with *Trichoderma* strains, slow growth was observed. Further radial growth stagnated, forming a distinct boundary between two different mycelial types (Figure 4). The three *Trichoderma* strains were almost equally efficient against H-Lt (Figure 5), with no significant difference based on Tukey’s test.

*F. siamensis* (H-Fs) is a member of *Polyporales*, a white rot fungus that has been found to infect a broad spectrum of urban trees in Singapore [6]. In culture, H-Fs produced slow growth of dense white mycelia, which gradually developed a yellow colour, and H-Fs was allowed to grow for seven days prior to *Trichoderma* inoculation. In the presence of *Trichoderma* strains, H-Fs growth was stagnated (Figure 4) with more prominent inhibition by Th-1 and Th-2, as indicated by Tukey’s analysis. Following mycelial intersection, *Trichoderma* mycelia from all three strains were observed to grow over H-Fs mycelia.

*G. orbiforme* (H-Go) was observed to display slow growth of branching white mycelia that grow deep into the PDA medium, forming folds in the agar surface, and was allowed to grow first for seven days (reaching a radical growth of 2.1 cm) before *Trichoderma* inoculation. Radial growth of H-Go in the control plate was 6.3 cm on Day 7 post-inoculation. Following *Trichoderma* inoculation, slower growth in H-Go mycelia could be observed for all *Trichoderma* strains. *Trichoderma* mycelia were observed to grow over H-Go mycelia, and H-Go mycelia near the interphase appeared to become elevated and thicker than in the control plate. Using Tukey’s test, inhibition by Th-1 on H-Go was significantly different from the inhibition by Th-2 and Th-3.

*G. australe* (H-Ga) is a white rot fungus that has been found to be a major wood decay fungus in ornamental palms and Casuarina trees [6]. H-Ga was observed to produce white mycelia that grew into the PDA medium, forming folds in the agar surface. Due to its slow growth, H-Ga was allowed to grow for 7 days (reaching a radial growth of 1.8 cm) prior to *Trichoderma* inoculation. Radial growth of H-Ga was 8.6 cm on Day 7 in the control plate. In Figure 4 and Figure 5, slower growth could be observed for H-Ga for all *Trichoderma* strains after inoculation. *Trichoderma* mycelia were observed to grow over H-Ga mycelia, and the production of a brown-pigmented interaction zone could be observed on the surface of H-Ga near the interphase. Using Tukey’s test, inhibition by Th-1 against H-Ga was similar to Th-2; both were significantly more efficient than the inhibition by Th-3.

Th-1 and Th-2 inhibit pathogenic fungal growth through volatile organic compounds (VOCs). Inhibition by a VOC test was conducted for H-Pn (*P. noxius*) by examining its growth after exposure to Th-1 and Th-2 volatiles without direct contact or growing in the same medium. (Figure 6). H-Pn’s susceptibility to inhibition by *T. harzianum* strains made it a good model for further mechanistic studies.

H-Pn mycelia covered the entire control plate by Day 6. When sharing the headspace with either *Trichoderma* strain (Th-1 or Th-2) but without direct contact or connection through the medium, growth rates of H-Pn were significantly lower. On Day 6, the radial growth diameters of H-Pn under the influence of volatiles from Th-1 and Th-2 were 3.2 ± 0.4 cm and 3.6 ± 0.4 cm, respectively, which were equivalent to inhibition percentages of 62.4% and 58.5%, respectively.

To test if such VOC inhibition was transient or permanent, the Th-1 plate was removed from the H-PN plate after seven days of sharing the headspace with H-Pn, where the growth rate was significantly inhibited (Figure 6C(2)). The H-Pn plate was assembled with a blank PDA plate on Day 7. It was observed that a thin layer of H-Pn mycelia was able to grow from the central area of dense mycelia. H-Pn mycelia were then observed to fill the entire plate within a single day (Day 8) (Figure 6C(3)). On the other hand, H-Pn plates under the continuous influence of Th-1 VOC continued slow growth, with H-Pn mycelia completely covering the plate only on Day 14.

### 3.4. Th-1 Releases Fungal Inhibitory Volatiles Phenylethyl Alcohol and Cadina-1,4-diene

An SPME GC-MS analysis (Figure 7) was conducted for the four samples taken from the VOC test, namely the Th-1 control (CE0), H-Pn control (PN0), Th-1 in the experiment plate (CE1), and H-Pn in the experiment plate (PN1) on Day 7. Each sample comprised six cut plugs of fungal mycelia. We aimed to know the identities of the volatiles, to check the presence of known antifungal volatiles, and understand possible volatile profile changes under the influence of volatiles from the other fungal species.

The GC chromatograms obtained for CE0 and CE1 appeared to be similar (Figure 7) with many common peaks. The predominant volatiles detected from Th-1 in a high-percentage area and analysis quality were phenylethyl alcohol (percentage area of 3.70% in CE0, 7.07% in CE1) and cadina-1,4-diene (percentage area of 8.13% in CE0, 9.92% in CE1). It was noted that these two volatiles were present at a higher percentage of total volatiles in CE1 than CE0. Several other volatiles shared between the two spectra were detected with a high-percentage area; however, the analysis quality was insufficient to infer the identity of these volatiles. Lists of the volatiles identified by SPME GC-MS in the four samples are provided as a Appendix A (Appendix A of GC-MS data).

The PN1 spectrum appeared to show more distinct peaks than PN0, suggesting changes in volatiles in response to *Trichoderma* VOCs. Most of these peaks corresponded to volatiles with low analysis quality, so no conclusion could be drawn regarding changes in the volatile profile.

### 3.5. Th-1 Exhibits Mycoparasitic Mycelial Interactions

Microscopy was conducted to observe the interactions between the mycelia of Th-1 and the six pathogenic fungal species to determine if direct physical contact between mycelia was a significant factor contributing to the inhibitory effects of *Trichoderma* species on pathogenic fungi and observe how the mycelia interact with each other (Figure 8).

It was observed that Th-1 mycelia had the tendency to coil around the hyphae of pathogenic fungi species, a possible form of mycoparasitic behaviour (Figure 8).

## 4. Discussion

### 4.1. CE92 (Th-1) Is the Best BCA Candidate

To screen for the best BCA candidates, the growth rate was first compared among the *T. harzianum* strains, with the assumption that rapid growth is a key contributor to the functionality of a BCA. *T. harzianum* was chosen as the focus of this project based on experiments conducted in our laboratory prior to this project, which suggested *T. harzianum* was the most promising BCA among all the *Trichoderma* species we isolated. A high growth rate enables the environment surrounding the trees to be more quickly colonised, reducing the chance of pathogenic fungi accessing wounded areas on the roots or other parts of a tree. After the growth rate comparison, we selected the two fastest-growing *T. harzianum* strains for further studies: Th-1 and Th-2. Th-3 was also selected for its fast growth and its distinctive morphology (Figure 3).

With dual culture analysis, all three strains of *T. harzianum* could inhibit the radial growth of all the tested pathogenic fungal species, although the extent of inhibition varied. Overall, Th-1 displayed the highest mean percentage of inhibition against all pathogenic fungi. Th-1 inhibition of *G. orbiforme* (H-Go) is more significant than Th-2 or Th-3. For *G. australe* (H-Ga), inhibition by Th-1 was similar to Th-2 but significantly more than Th-3.

The fast-growing pathogenic fungi seem to be more effectively inhibited by the *Trichoderma* species with >50% inhibition for *P. noxius* (H-Pn), *R. microporus* (H-Rm), and *L. theobromae* (H-Lt), suggesting that they are most susceptible to inhibition by *Trichoderma* species. The best inhibition was achieved for *P. noxius*, with inhibition exceeding 80% for both Th-1 and Th-2. The inhibition against the three slower-growing fungi was less effective (inhibition <50%), with *G. orbiforme* (H-Go) being the least susceptible to *Trichoderma* inhibition.

We also conducted a cut plug assay to reinforce the results of the dual culture assay. It is a possible stricter test of BCA’s potential to combat phytopathogenic fungi by providing a more challenging environment for the BCA to exhibit its inhibitory effect. The cut plug assay proved Th-1 to be the most potent BCA candidate, causing the formation of inhibition rings after placing some pathonogenic fungi onto fully colonised plates. It was the most potent against *Phellinus noxius* (H-Pn), with a large inhibition ring formation and possibly stressing and killing *P. noxius* (H-PN) hyphae, as suggested by the brown pigmentation and the thinning of white mycelia (Appendix A).

Future work could consider testing Th-1 on infected wood blocks, which would further validate Th-1’s ability to treat infected trees. However, it is likely that the *Trichoderma* isolates, including Th-1, would function more optimally to prevent rather than to cure phytopathogenic fungal infection of healthy trees. This prediction is based on observations in the dual culture assay, where interaction between the *Trichoderma* and pathogenic fungal mycelia led to the formation of a clear boundary.

It may be useful to test if pathogenic fungal mycelia that have been mounted over Th-1 mycelia in dual culture studies are still alive. This could be conducted by cutting plugs at that location and plating them onto PDA plates. If only the *Trichoderma* strain can be revived, it could suggest that the pathogenic fungi are killed by the *Trichoderma*. Alternatively, microscopic methods using dyes that selectively stain dead cells could be employed [30]. If such tests show significant levels of death in pathogenic fungi, they may suggest that Th-1 can kill pathogenic fungi, which would boost its potential as a BCA.

Th-1 appears to be the most promising biocontrol agent (BCA) candidate among the tested *T. harzianum* strains given its ability to inhibit all six pathogenic fungi while also being one of the fastest-growing strains.

### 4.2. Mechanistic Studies

The mechanisms employed by BCAs are many and complex, which are not only variable among species and strains but are also influenced by factors such as soil type, temperature, pH, and the presence of other microflora [31]. Being the most susceptible to inhibition by *T. harzianum* (Th-1) and *P. noxius*, H-Pn was identified as the ideal pathogenic fungal species to serve as a model for testing the mechanism in which *Trichoderma* species could exert their inhibitory effects. Therefore, the VOC test, filtrate test, and microscopic analysis were designed to determine if Th-1 mainly inhibited pathogenic fungal growth through the release of volatiles, secretion of soluble metabolites, or direct contact between mycelia, respectively.

The production of volatiles by *Trichoderma* species as a potential means of biocontrol has been studied since the 1970s [24]. Our VOC test results affirmed that volatiles play a key role in Th-1’s inhibition of pathogenic fungi growth, which was similar to a previous report [32] that found that VOCs produced by *Trichoderma asperelloides* PSU-P1 were inhibitory to fungal pathogens. These volatiles were also found to activate defence responses and promote plant growth in *Arabidopsis thaliana*. A few chemical compounds were tentatively identified by SPME GC-MS. Our work further identified the transient nature of the inhibition, where rapid growth of H-Pn mycelia resumed following the replacement of the Th-1 plate with a blank PDA plate, suggesting that the presence of volatiles emitted by *Trichoderma* species could influence H-Pn growth. Further analysis of the volatiles via SPME GC-MS identified phenylethyl alcohol and cadina-1,4-diene as two major volatile chemicals produced by Th-1. Other studies have shown that phenylethyl alcohol, which was isolated from other *Trichoderma* species, exhibits antifungal activity [14,33], while cadina-1,4-diene is known to be a major component of various antifungal essential oils [34,35]. In a study of the antifungal effects of phenylethyl alcohol in comparison with common antifungal drugs, phenylethyl alcohol was found not only inhibitory to the human pathogenic fungus *Candida albicans* by itself but it also significantly increased the antifungal effects of common antifungal drugs [36]. Therefore, phenylethyl alcohol and cadina-1,4-diene could be the key contributors to the antifungal activities of Th-1 volatiles. They could either act additively or synergistically. While Phenylethyl alcohol has been previously reported in *T. harzianum* [37], cadina-1,4-diene has not been reported in any *T. harzianum*, although it has been reported to be emitted by *T. reesei* [38].

Volatile profiles for Th-1 incubated alone and connected with H-Pn through headspace were similar, with higher relative contents of phenylethyl alcohol and cadina-1,4-diene, suggesting that the emission of the antifungal volatiles was constitutive but can be enhanced by the presence of a target pathogenic fungus. This characteristic would allow Th-1 to serve as a stable BCA by continuously producing antifungal volatiles to inhibit pathogenic fungal growth in the environment. The upregulation of phenylethyl alcohol and cadina-1,4-diene production in the presence of H-Pn also suggests that Th-1 can sense pathogenic fungi in the environment and enhance its antifungal activity as a response.

We also tested the filtrates of the Th-1 culture but failed to find any significant impact on H-Pn growth. This finding is different from those reports demonstrating significant inhibitory activity of *Trichoderma* filtrates against fungal pathogens [39]. It may be necessary to compare our protocol with those used by others and fine-tune the protocol accordingly. It is also possible that the secreted metabolites selectively inhibit certain species of pathogenic fungi, and this mechanism is not applicable to this Th-1/H-Pn combination. Future studies could conduct more filtrate tests for more pathogenic fungal species/*Trichoderma* strain combinations. The concentration of metabolites in the filtrate may also have been too low. An alternative method employed by some papers is the chemical extraction of the culture filtrate to obtain a crude extract, which can be diluted as required [40]. This would also allow the identification of specific compounds present in the filtrate that have antifungal potential.

When soluble extract exhibited little inhibition activity, inhibition may be mediated through direct contact (Figure 4 and Figure 6). Microscopic analysis suggests that Th-1 hyphae coiled around the hyphae of the pathogenic fungi, a reported mycoparasitic behaviour [41,42]. These two reports also identified other mechanisms of mycoparasitism, such as the formation of appressorium and the lysis of pathogenic fungal mycelia through hydrolytic enzymes, which could not be identified in the current study. The curling of hyphae has been linked to mycoparasitism, and the close contact created has been suggested to facilitate the disruption of pathogenic hyphae by the endo- and exo-chitinase. *Trichoderma*-produced proteases could also effectively inactivate the hydrolytic enzymes produced by the pathogen [31].

One challenge encountered during microscopic analysis was the different growth rates of the fungal species, which made it hard to estimate when the two types of mycelia would intersect. Furthermore, the progression of any mycoparasitic characteristics could not be observed after staining as the lactophenol blue stain kills the fungal cells. Thirdly, similar-sized and look-alike hyphae also made it difficult to draw strong conclusions for several *Trichoderma*/pathogenic fungi combinations. Future studies could consider the inoculation of fungal hyphal tips onto the PDA-coated microscope slides at staggered intervals. This would increase the chances of visualising the point where mycelial interactions are just beginning to occur as well as mycoparasitic features, such as the progressive coiling of the mycelia with time. This may also allow the lysis of pathogenic fungal hyphae to be observed and would give a deeper understanding of the mycoparasitic features of Th-1.

## 5. Conclusions

In conclusion, this study identified and demonstrated that a local *Trichoderma harzianum* strain CE92 (Th-1) holds good potential as a BCA against a broad spectrum of pathogenic wood rot fungi that could potentially infect multiple species of urban trees in Singapore. It has the advantages of fast growth and broad-spectrum in vitro activities against multiple phytopathogenic fungal species. Furthermore, the mechanistic studies suggest that its inhibition activities were mediated by both emitted volatiles and direct mycelial contact.

## Figures and Tables

**Figure 1 jof-09-00675-f001:**
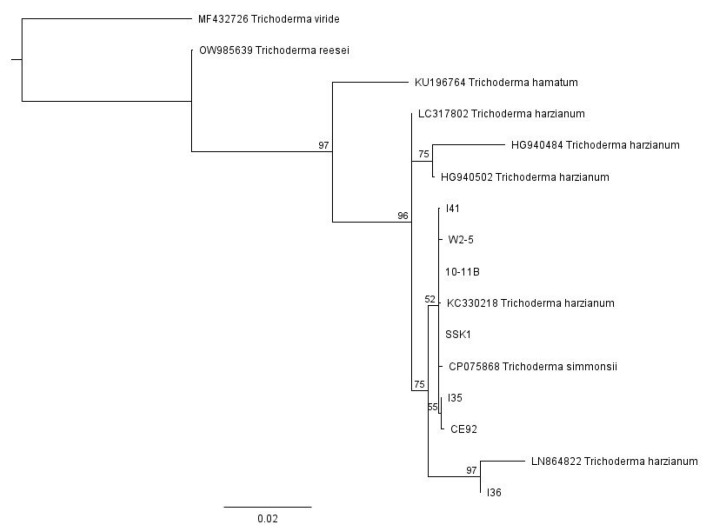
The neighbour-joining phylogenetic tree of the *Trichoderma* isolates with 100 bootstrap values.

**Figure 2 jof-09-00675-f002:**
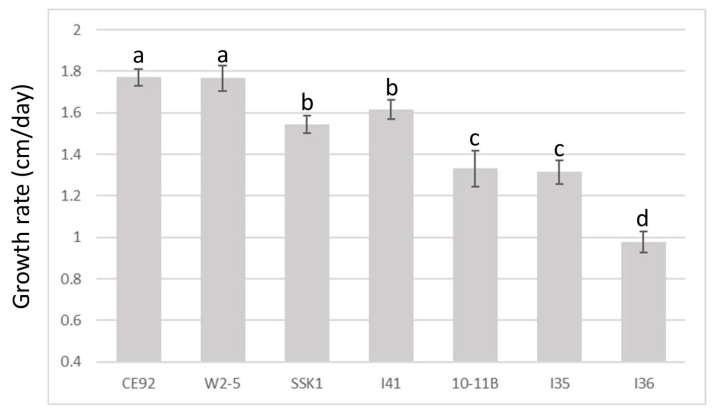
Comparison of growth rates of the seven *T. harzianum* strains. A cut plug was used to inoculate a new plate and the radial growth of hyphae was measured daily. a, b, c, and d are the rankings by Tukey’s test with a 95% significant difference (*p* < 0.05) from each other; error bars denote standard deviations (n = 5).

**Figure 3 jof-09-00675-f003:**
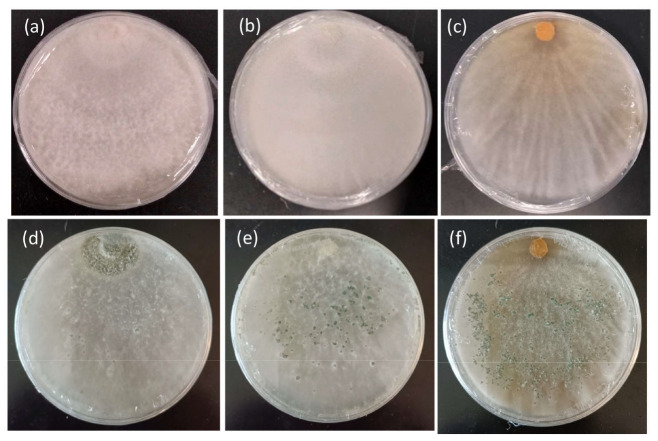
Morphology features of the mycelial growth of three *T. harzianum* strains on PDA. (**a**) Th-1 (CE92, (**b**) Th-2 (SSK1), and (**c**) Th-3 (W2-5) after 7 days of culture. (**d**–**f**) are the same strains sporulating after 3 weeks of culture.

**Figure 4 jof-09-00675-f004:**
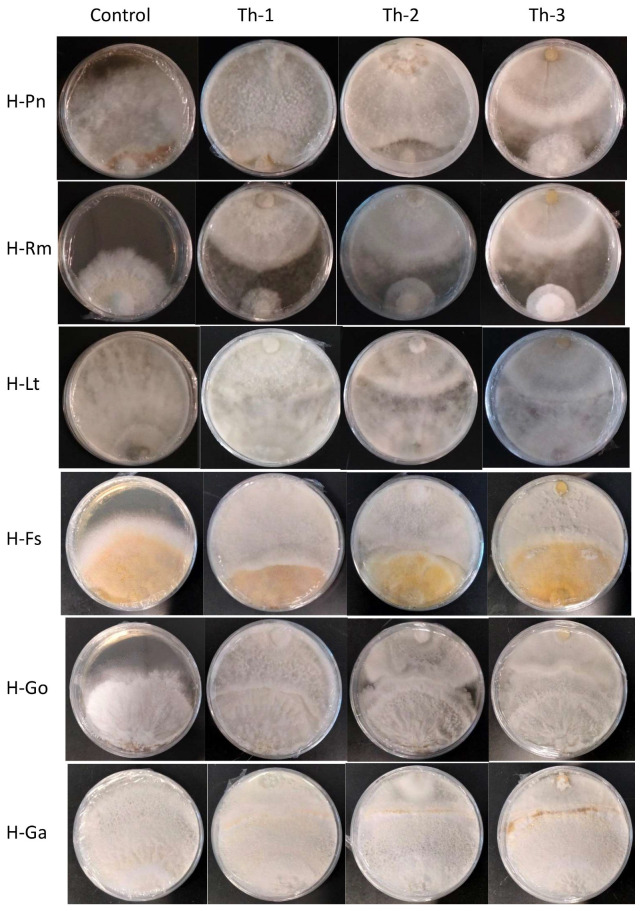
The representative plates for dual culture assays seven days after placing the *T. harzianum* plugs. Control plates had only pathogenic fungal growth without *Trichoderma* plugs. Th-1, Th-2, and Th-3 were the *T. harzianum strains*, and H-Pn, H-Rm, H-Lt, H-FS, H-Go, and H-Ga were the six pathogenic fungi. Plugs for the pathogenic fungi were placed at the bottom and *Trichoderma* plugs were placed at the top of each plate. For the fast-growing pathogenic fungal strains (H-Pn, H-Rm, and H-Lt), a *Trichoderma* plug was placed at the same time as a pathogenic fungal plug. For the slow-growing pathogenic fungal strains (H-Fs, H-Go, and H-Ga), their plugs were placed on the plate and allowed to grow for seven days before placing the *Trichoderma* plug.

**Figure 5 jof-09-00675-f005:**
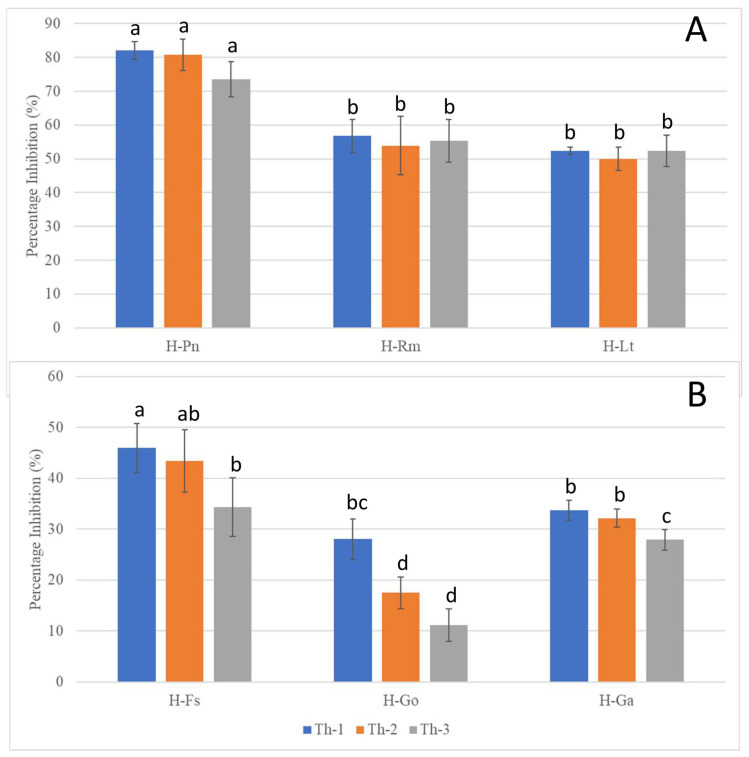
Inhibition percentages of the pathogenic fungi in dual culture assays seven days after *T. harzianum* strain inoculation. (**A**) Inhibition of the fast-growing pathogenic fungi (H-Pn: *P. noxius*; H-Rm: *R. microporus*; H-Lt: *L. theobromae*) with same-time inoculation of *Trichoderma* strains. (**B**) Inhibition of the slow-growing pathogenic fungi (H-Fs: *F. siamensis*; H-Go: *G. orbiforme*; H-Ga: *G. australe*, which were allowed to grow for seven days prior to *Trichoderma* inoculation. a, b, c, and d are the rankings by Tukey’s test with a 95% significant difference (*p* < 0.05) from each other; error bars denote standard deviations (n = 3).

**Figure 6 jof-09-00675-f006:**
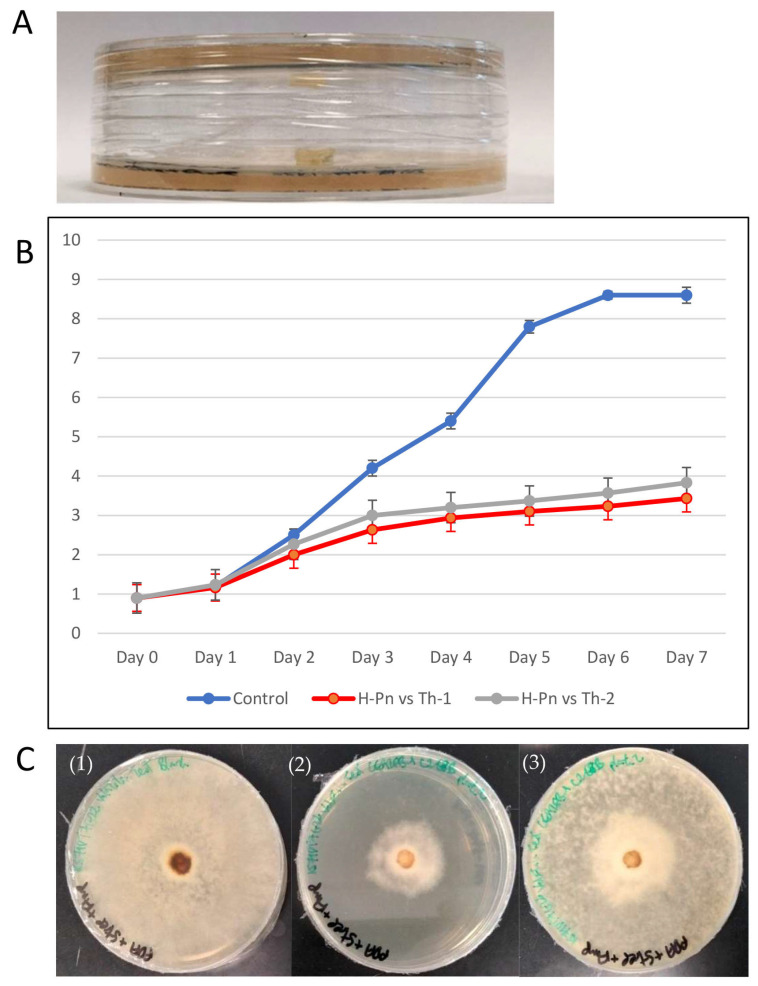
*Trichoderma* VOC inhibition of *P. noxius* growth. (**A**) The experiment setup with the plate inoculated with a *Trichoderma* plug facing down over another plate inoculated with an H-Pn plug that was sealed. (**B**) Radial growth of H-Pn mycelia without the *Trichoderma* plate above (control) and in the presence of Th-1 or Th-2 plate 1-7 days after assembly (n = 3). (**C**) Test of transient inhibition by VOC. (1) H-PN with no *Trichoderma* exposure (control), seven days after the assembly; (2) the H-PN plate seven days after the assembly; (3) the Th-1 plate was removed, and the H-Pn plate was covered and allowed to grow for one more day.

**Figure 7 jof-09-00675-f007:**
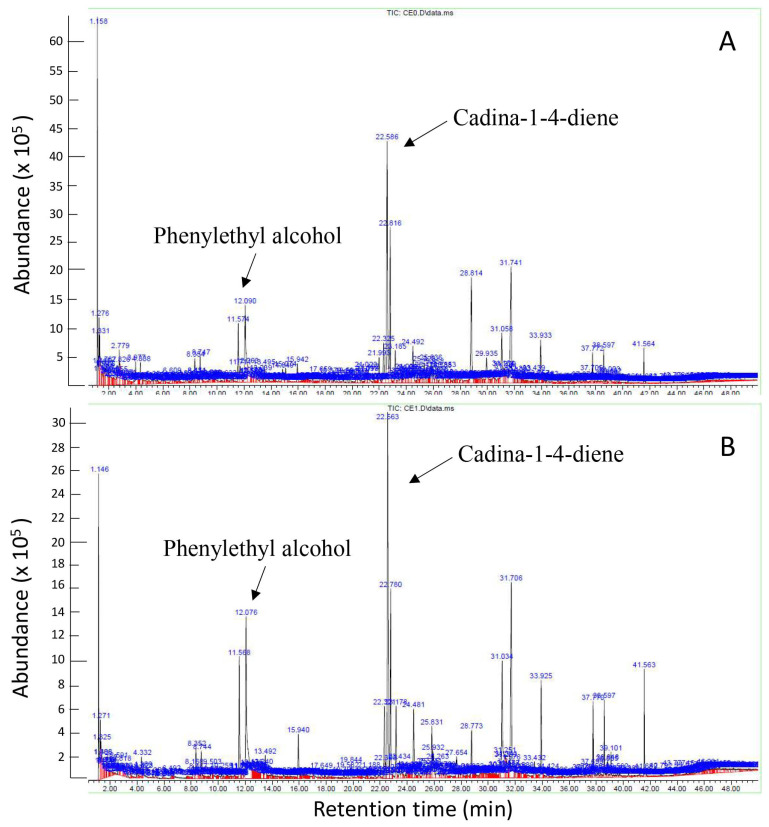
GC chromatograms for Th-1 (*T. harzianum*) volatiles (CE0, (**A**)) and TH-1 volatiles after sharing the headspace with H-Pn (*P. noxius*) for 7 days (CE1, (**B**)).

**Figure 8 jof-09-00675-f008:**
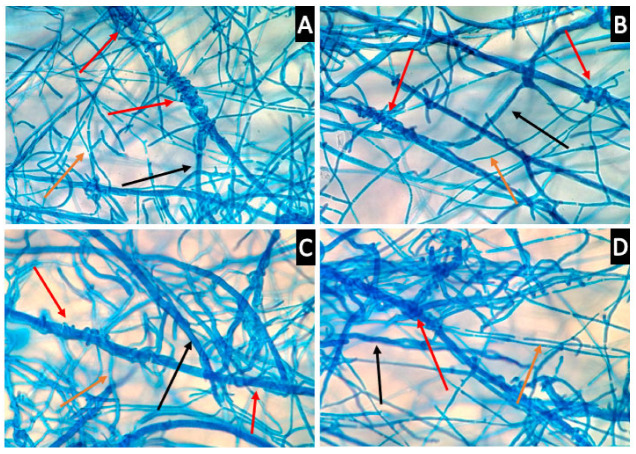
*Trichoderma* hyphae coil around the hyphae of target pathogenic fungi. (**A**,**B**) coiling of Th-1 (*T. harzianum)* hyphae around H-Pn (*P. noxius)* hyphae (400× magnification, red arrow). The bigger size hyphae belonged to H-Pn, black arrow) and the smaller size hyphae belonged to Th-1 (orange arrow). (**C**,**D)** coiling of Th-1 hyphae around *L. theobromae* (H-Lt) hyphae (400× magnification, red arrow). The bigger size hyphae belonged to H-Lt (black arrow) and the smaller size hyphae belonged to Th-1 (orange arrow).

**Table 1 jof-09-00675-t001:** Ingredients for PDA culture media (800 mL).

Components	Quantities Added
0.03 g/mL Streptomycin	800 µL
0.1 g/mL Ampicillin	800 µL
MilliQ Water	800 mL
Potato Dextrose Agar Powder	31.20 g

**Table 2 jof-09-00675-t002:** List of pathogenic fungi and *Trichoderma* strains.

Sample ID (Abbreviation, Code)	Molecular Identify	GenBank Accession Number	Isolation Source
CE92FB_1C2G6B (CE92, Th-1)	*Trichoderma harzianum*	OQ789693	*A Fulvifomes siamensis* fruiting body collected from a *Casuarina equisetifolia* tree, West Singapore
SSK1-A12AG2 (SSK1, Th-2)	*Trichoderma harzianum*	OQ789697	A *Ganoderma* sp. fruiting body collected from an unidentified tree in Singapore Botanic Gardens
W2-5G2 (W2-5, Th-3)	*Trichoderma harzianum*	OQ789699	A *Ganoderma boninense* fruiting body collected from a MacArthur Palm (*Ptychosperma macarthurii*), Central Singapore
10-11BBG2 (10-11B)	*Trichoderma harzianum*	OQ789692	A *Ganoderma boninense* fruiting body collected from a sealing wax palm (*Cyrtostachys renda*), Central Singapore
I36G2B (I36)	*Trichoderma harzianum*	OQ789695	Soil of *Brassica rapa* cv. Caixin, West Singapore
I35G2 (I35)	*Trichoderma harzianum*	OQ789694	Soil of *Brassica rapa* cv. Caixin, West Singapore
I41G3 (I41)	*Trichoderma harzianum*	OQ789696	Soil of *Amaranthus tricolor*, West Singapore
KS71DT7G14 (H-Pn)	*Phellinus noxius*	OQ558864	Khaya Tree (*Khaya senegalensis*) diseased tissue, West Singapore
YF37#2AG29 (H-Fs)	*Fulvifomes siamensis*	OQ618213	A fruiting body on a Yellow Flame tree (*Peltophorum pterocarpum*), Central Singapore
PM162DT1-2G4 (H-Lt)	*Lasiodiplodia theobromae*	OQ558857	Diseased tissue collected from a *Purple Millettia* tree (*Callerya atropurpurea*), Central Singapore
P167FBAG6 (H-Go)	*Ganoderma orbiforme*	OQ558851	A fruiting body collected from a foxtail palm (*Wodyetia bifurcate*), Central Singapore
P172FB2G1 (H-Ga)	*Ganoderma australe*	OQ558853	A fruiting body collected from an Areca palm (*Dypsis lutescens*), West Singapore
P175FB2G2 (H-Rm)	*Rigidoporus microporus*	OQ558869	A fruiting body collected from a cabbage palm (*Roystonea oleracea)*, West Singapore

## Data Availability

Detailed data can be available upon request.

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
