# Peer review of "Locally Isolated Trichoderma harzianum Species Have Broad Spectrum Biocontrol Activities against the Wood Rot Fungal Species through Both Volatile Inhibition and Mycoparasitism"

_jof, 2023, doi:10.3390/jof9060675_

Round 1

Reviewer 1 Report

Summary

In the manuscript the authors assessed the in vitro biocontrol activity of seven locally isolated Trichoderma species against six major wood rot fungal species (Basidiomycota and Ascomycota) that had all been isolated from trees and soil in the area of Singapore. The evaluation of the potential of local Trichoderma species to combat root rot fungi led to the identification of one Trichoderma harzianum isolate with high in vitro potential as BCA.

General concept comments

The article seems to represent preliminary work for identifying and characterizing Trichoderma biocontrol agents from tree habitats (fruiting bodies of plant pathogens on infected areas and soil) and is a follow up to a recently published article on the identification of several plant pathogens, published in Journal of Fungi in April 2023 (https://pubmed.ncbi.nlm.nih.gov/37108914/).

The authors first assessed growth rate of all Trichoderma spp. and then tested three Trichoderma spp. in dual culture assays to investigate their mycoparasitic potential. The best Trichoderma candiate was used in three additional tests to further characterize the mycoparasitic capabilities against a one to two root rot fungi. While inhibition by soluble secreted small molecules could not be detected against the tested root rot fungus, inhibition by volatile organic compounds was highly effective. They identified two VOCs by SPME GC-MS as most promising organic compounds. Using light microscopy they studied coiling of T. harzianum around pathogenic fungal hyphae of two hosts.

The article feels in parts like being copied from a work report. The section headings do not describe the section but only state the used method eg section title “cut plug assay”. In several sections the rationale behind the used method or the chosen strains is not clear. Although the authors explain, how they selected the T. harzianum isolate for the final characterization in the results, it is not completely clear, how the plant pathogens (1-2) were selected and why they did not include all plant pathogens in these three final tests, because this would be relevant, if this Trichoderma species should be used as BCA.

A lot of references are missing throughout the manuscript, to show, why assays were chosen and what is known in the literature about related Trichoderma spp. and their biocontrol activity.

The authors should take more care for figure preparations. The diagrams seem to be copy-pasted directly from excel or the analysis software for GC-MS analysis, which impacts negatively their quality and in some parts data points and standard errors are missing, or standard errors seem to be wrongly assigned.

The authors chose a very complicated nomenclature system. I would suggest that the nomenclature for all Trichoderma harzianum strains and plant pathogens should be reassessed and maintained so that it is easier to follow without having to got back to the first section and find the species name. Eg. you could use the abbreviation “Th” for Trichoderma harzianum and numbers from “1”-“7” for the T. harzianum isolates and “H-“ for plant pathogenic hosts followed by their initials referring to the species eg Phellinus noxius: “H-Pn”

Except for the “cut plug” assay the methods are established within the Trichoderma community for characterizing the in vitro mycoparasitic activity. For the “cut plug” assay no mention of a previous establishment of this method was made, it is not backed with a citation, and I honestly don’t understand, how it should work and how the authors measured mycoparasitic activity here. It seems they only visually assessed if Trichoderma can grow on a plate fully overgrown by a plant pathogen.

I´m not convinced by the GC-MS analysis, if one looks at the abundance axis, the two indicated VOCs are not more strongly accumulated in figure 8 B than in figure 8A.

Discussion: I don’t see a lot of discussion in the discussion section. Most of it is only summarized results again. Only the last paragraph contains a little more discussion.

Suggested content that is missing in the discussion:

·         Do you have any explanation, why you only isolated Trichoderma harzianum species from your samples. Is it maybe due to the sampling method, or do you have any suggestions in the literature for that?

·         What are the two compounds that you identified in GS-MS (phenethyl alcohol and cadina-1,4-diene). Do you think they have potential in BCA and why? Where they identified earlier in Trichoderma spp. or is this a new finding?

Specific comments

Line 67-68: The citations are one-sided

Line 71: Citation for prior studies are missing

150 Only duplicates for growth rate?

154 I suggest to rename it to “Dual Culture Assay” instead of “Dual Culture inhibiting Assay”

166 “Cut plug assay”? What is this exactly? Did you develop this method yourself? I never heard of it. Please use citation if you adapted this protocol from somewhere or explain how the assay works in high detail, if you developed it yourself.

214 How do you inoculate hyphal tips?

230: Please summarize your findings in a concise title for the section

231 Please provide detailed sampling information (in the manuscript and detailed supplementary file) Please provide a description of the source of Trichoderma species, how they were isolated and from which habitat or tissue and a clear labelling throughout the whole manuscript if they were isolated from soil or fruiting bodies of pathogens.

252 There is no mention of the four T. harzianum species that were chosen for the phylogeny and why they were used as reference. Please describe and explain.

254 How closely are the isolates related and are you sure, that some or all are not actually the same? Supplementary information on the sequencing results are missing.

258: Please summarize your findings in a concise title for the section

Please provide a table with calculated growth rate of the seven Trichoderma species and also assess the production of conidia by all species. This is important, if Trichoderma should be used as BCA.

Figure 2: This is not a comparison of growth rate but radial growth of 9 days. Statistics are missing for the growth assay. Also the experiment was only performed in duplicates. The experiment should be repeated to confirm the results. There are two “dots” at the end of the sentence in the figure legend.

Figure 3: Please show additional pictures of fully sporulated plates (need to be grown in light/dark cycle), since this is a major trait that also helps distinguish the species. (c) looks like there were two agar plugs inoculated one on the upper and one on the lower side of the plate. Why?

275: Please summarize your findings in a concise title for the section

289: omit extra title/section, this section belongs to the section starting in line 275

326: What is the ranking? Is “a” better or “d”? Please explain here or in the methods section

W2-5 has a slower growth rate by nearly 2 days slower. This has to be taken into consideration for inhibition of growth in the dual confrontation assays

368 Please summarize your findings in a concise title for the section

369-376: Please explain the idea behind the cut plug assay briefly. Even though I have several years of experience with Trichoderma culturing and biocontrol, I never heard of it, and don’t understand what it is good for, or how it is conducted.

Figure 6: please indicate in one picture what was measured. CK is not indicated but Control! I don’t see strong differences between the photos of one row, except for the KS71 row

401: Please summarize your findings in a concise title for the section

Why was only one strain tested in this assay. Please explain in this section.

Figure 7A please indicate which strain was on top and which on the bottom

7C: Why is there no standard error for the control, and the standard error for the other two samples look, like they are off! I think there is something wrong with the excel graphics! Please check thoroughly your diagrams, when you arrange them in the manuscript.

428: Please summarize your findings in a concise title for the section

429-432: It is not clear from the initial sentences, what exactly was analysed here. Please explain more precisely. Please explain in the GC-MS analysis why you think that these two compounds differ between CE0 and CE1. To me it looks like the two indicated VOCs are not more strongly accumulated in figure 8 B than in figure 8A. The abundance axis are difficult to read in Figure 8, but the scale is not the same, so Figure 8 B looks like there is more, even though that is not true. Please adapt scale in both diagrams, so that they are comparable.

443-445: please provide in a supplementary file the data for the detection of volatiles.

446-449: Please show the spectrum for PN samples at least in a supplementary file.

450: Please summarize your findings in a concise title for the section.

451-458: This test might not be suitable for testing the secretory potential of Trichoderma spp. Normally another test is conducted where PDA plates are covered with cellophane and a agar plug of a fresh Trichoderma (antagonist) colony is placed in the middle of the plate on top of the cellophane. After 2 days of growth Trichoderma is removed with the cellophane and the host (plant pathogen is placed in the middle of the plate and growth is recorded. The cellophane allows for passage of small molecules and secreted enzymes. See also: Dennis and Webster 1971 https://doi.org/10.1016/s0007-1536(71)80077-3 and Reithner et al., 2005 https://doi.org/10.1016/j.fgb.2005.04.009

I would suggest to repeat the analysis using the assay setup described in the publications above and see if antagonistic growth is found here.

459: Please summarize your findings in a concise title for the section

Figure 9: white arrows are poorly visible. Please indicate the magnification in the pictures. Pictures are messy. Indicate that red arrows show coiling.

480: It is not clear, why you selected the strains, please explain what were the major results that led you to use this T. harzianum strain.

Moreover: Do you have any explanation, why you only isolated Trichoderma harzianum species from your samples. Is it maybe due to the sampling method, or do you have any suggestions in the literature for that?

489-490: The assay for fast growing fungi and slow growing fungi cannot be compared directly, since due to their different growth characteristics and the assay setup (they are allowed to grow on the plate for several days without the antagonist Trichoderma) the conditions are not completely comparable. Please take this into account for the discussion.

506-508: Please explain why based on your findings you think Trichoderma would function more optimally to prevent infection rather than to cure trees completely.

509-510: I doubt that it is possible to separate the hyphae of the two fungi for this test.

526: This is not the only mention of VOCs produced by Trichoderma. Studies on VOCs date back to the 1970s. Please correctly cite the evidence for VOCs production by Trichoderma in relation to biocontrol.

536-543: What are the two compounds that you identified in GS-MS (phenethyl alcohol and cadina-1,4-diene). Do you think they have potential in BCA and why? Were they identified earlier in Trichoderma spp. or is this a new finding?

555-560: This is not a new finding. Curling around hyphae has been identified in many Trichoderma species before. Please provide references where curling around hyphae has been identified previously. What is the benefit for the fungus to curl around hyphae of hosts? Please provide references for that.

592: The authors must provide data. Data should be made publicly available. Sequencing data and detailed analysis data from the VOCs tests etc. must be made available in supplementary files or a public repository.

Only minor editing of English language required.

Author Response

Dear reviewer, we want to thank you for the many critical but helpful comments and suggestions. We have conducted a major revision to the manuscript by adding more data, remaking some figures, and including more relevant references on top of the editorial changes as suggested. You can find our detailed point-to-point responses in the attached file. Thanks again for helping us improve our manuscript!

Reviewer 2 Report

This article provides useful information of “Locally isolated Trichoderma species have broad spectrum biocontrol activities against the wood rot fungal species through both volatile inhibition and mycoparasitism”. The scope and results are very clear. See the comment in the MS.

Author Response

(The authors gave the same response as above.)

Reviewer 3 Report

This MS provides insightful information about the potential of local Trichoderma strains as biocontrol agents for wood rot fungal species that infect multiple urban tree species in Singapore. The authors have assessed the Trichoderma strains by DNA barcoding for their molecular identities and tested them for their growth rate and effectiveness in inhibiting the pathogenic fungi using in vitro dual culture assay and cut-plug assay. The results indicate that Trichoderma harzianum strain CE92 was the most effective in inhibiting the growth of the pathogenic fungi tested. The study also suggests that volatile organic compounds production and direct hyphal contact contributed to inhibition. The identification of known fungal inhibitory volatiles using SPME GC-MS adds value to the study. The observation of Trichoderma harzianum strain CE92 coiling around Phellinus noxius and Lasiodiplodia theobromae upon contact in vitro provides a clue to the possibility of mycoparasitism.The authors have presented their findings clearly and concisely. The study provides valuable information about Trichoderma inhibition of pathogenic fungi and identifies local strains with good potential as broad-spectrum biocontrol agents against root/wood rot fungi in Singapore. However, some of the observations made in this study need to be validated through further experiments. Overall, this is a well-conducted study, andI recommend addressing the following concerns before its publication in the journal:

1.The Latin names of fungi should be abbreviated when appearing for the second time, and tables should be presented in four-line, three-box format without vertical lines.

2. No conclusion could be drawn based on the volatiles assay. Why not conduct a independent assay to test if phenylethyl alcohol and cadina-1,4-diene play a role in the inhibition of fungal growth? Appearance in common in both CE0 and CE1 does not mean they are the effective compound.

Author Response

(The authors gave the same response as above.)

Round 2

Author Response

Dear reviewer, we appreciate very much for your further suggestions. We have made further revisions to our manuscript. Please find question specific elaborations in the attached file.
